# Long-Term Administration of *Vespa velutina nigrithorax* Venom Ameliorates Alzheimer’s Phenotypes in 5xFAD Transgenic Mice

**DOI:** 10.3390/toxins15030203

**Published:** 2023-03-06

**Authors:** Yoon Ah Jeong, Hyun Seok Yun, Yoonsu Kim, Chan Ho Jang, Ji Sun Lim, Hyo Jung Kim, Moon Bo Choi, Jae Woo Jung, Jisun Oh, Jong-Sang Kim

**Affiliations:** 1School of Food Science and Biotechnology, Kyungpook National University, Daegu 41566, Republic of Korea; 2Korean Medicine Material Development Center, National Institute for Korean Medicine Development, Gyeongsan 38540, Republic of Korea; 3Department of Integrative Biology, Kyungpook National University, Daegu 41566, Republic of Korea; 4Institute of Agricultural Science and Technology, Kyungpook National University, Daegu 41566, Republic of Korea; 5Department of Nuclear Medicine, Keimyung University Dongsan Hospital, Daegu 42601, Republic of Korea; 6Institute of Plant Medicine, Kyungpook National University, Daegu 41566, Republic of Korea; 7External Herbal Dispensaries (BEEPLUS), Wonjae Oriental Medicine Clinic, Chilgok-gun 718807, Gyeongsangbuk-do, Republic of Korea; 8New Drug Development Center, Daegu-Gyeongbuk Medical Innovation Foundation, Daegu 41061, Republic of Korea

**Keywords:** wasp venom, *Vespa velutina nigrithorax*, Alzheimer’s disease, 5xFAD mouse, memory improvement, anti-inflammation

## Abstract

Alzheimer’s disease (AD), the most prevalent neurodegenerative disease, is characterized by progressive and irreversible impairment of cognitive functions. However, its etiology is poorly understood, and therapeutic interventions are limited. Our preliminary study revealed that wasp venom (WV) from *Vespa velutina nigrithorax* can prevent lipopolysaccharide-induced inflammatory signaling, which is strongly implicated in AD pathogenesis. Therefore, we examined whether WV administration can ameliorate major AD phenotypes in the 5xFAD transgenic mouse model. Adult 5xFAD transgenic mice (6.5 months of age) were treated with WV by intraperitoneal injection at 250 or 400 μg/kg body weight once weekly for 14 consecutive weeks. This administration regimen improved procedural, spatial, and working memory deficits as assessed by the passive avoidance, Morris water maze, and Y-maze tasks, respectively. It also attenuated histological damage and amyloid-beta plaque formation in the hippocampal region and decreased expression levels of pro-inflammatory factors in the hippocampus and cerebrum, while it reduced oxidative stress markers (malondialdehyde in the brain and liver and 8-hydroxy-2′-deoxyguanosine in the plasma). Overall, these findings suggest that long-term administration of WV may alleviate AD-related symptoms and pathological phenotypes.

## 1. Introduction

Neuroinflammation is a major contributor to the pathogenesis of neurodegenerative disorders, including Alzheimer’s disease (AD) [1,2,3,4,5,6]. We recently reported that wasp venom (WV) isolated from the invasive yellow-legged hornet *Vespa velutina nigrithorax* suppressed the activation of microglial cells in response to liposaccharide (LPS) treatment [7]. Moreover, intraperitoneal administration of WV to LPS-treated wild-type (WT) C57BL/6J mice improved neuroinflammation-induced memory deficits (Appendix A). While bee venom (BV) and its components, including melittin, have been used for centuries in traditional oriental medicine, there are few studies on the beneficial effects of WV.

Vespid wasps, which belong to the family Vespidae, are distributed worldwide and comprise more than 5000 species [8]. The components in WV are known to differ from those found in BV. WV is composed of multiple amines (such as histamine, tyramine, and serotonin), small peptides (such as anoplin, decoralin, eumenitin, eumenitin-R, EpVP, mastoparan, and rumenitin-F), and proteins (such as hyaluronidase, α-glucosidase, phosphatase, phospholipase A2, and phospholipase B) [9]. Certain components found in WV exhibit antimicrobial, anti-cancer, and anti-inflammatory activities [8,10,11]. The mastoparans are a group of small peptides isolated from the venoms of various wasps including *Vespula lewisii*, *Vespula vulgaris*, *Vespa crabro*, and *Polistes jadwigae.* Mastoparan-A, in particular, is characterized by its antitumor activity against melanoma cells and anti-microbial activity, although mastoparan-B and -M have been reported to have powerful hemolytic activity secondary to the stimulation of histamine release. Polisteskinin-R, produced by *Polistes rothneyi*, exerts potent anxiolytic effects [8]. In addition, phospholipase A2 from BV, also found in WV, was reported to improve AD-like pathological phenotypes in the 3xTg AD mouse model, presumably by increasing the regulatory T cell (Treg) count and thereby suppressing neurotoxic inflammatory signaling [12,13]. 

Contrary to these health-beneficial effects of the venoms, WV from *Vespa velutina nigrithorax* is one of the most prevalent causes of anaphylaxis in several Asian and European countries, including South Korea and Spain, posing a serious public health risk [14,15,16,17]. As a result, there has been a rise in demand among medical professionals and researchers for identification of WV components for both diagnostic and therapeutic purposes [15].

Although its etiology is not completely understood, AD is characterized by the irreversible decline of memory and cognition abilities along with three core pathologies: the accumulation of extracellular amyloid-beta (Aβ) plaque, the deposition of an intracellular neurofibrillary tangle of hyperphosphorylated tau, and a sustained immune response. Recent studies have shown that the neuroinflammation accompanying the persistent activation of microglia and astrocytes in the brain plays a major role in AD pathogenesis [18].

Numerous transgenic (Tg) animal models have been developed that recapitulate certain core features of AD, such as 5xFAD, Tg2576, APP23, APP/PS1, and 3xTg mice. The 5xFAD Tg model overexpresses human Aβ precursor protein (APP) harboring three AD-linked mutations [K670N/M671L (Swedish), I716V (Florida), and V717I (London)] and human presenilin-1 harboring two AD-linked mutations [M146L and L286V]. This model exhibits progressive histopathological and cognitive symptoms resembling those of AD [19]. While no Tg model perfectly replicates human AD [20], the 5xFAD model is advantageous due to early symptom onset, which saves on animal care costs.

Based on our previous studies showing that WV (isolated from *Vespa velutina nigrithorax*) suppressed activation of BV2 microglial cells [7] and improved cognitive deficits caused by LPS in mice (Appendix A), we conducted this study to examine whether WV can protect against the development of AD-like phenotypes in 5xFAD Tg mice.

## 2. Results

### 2.1. WV Treatment Improves Learning and Memory Function in 5xFAD Tg Mice

The 5xFAD Tg mice received WV, BV, donepezil (DONE), or vehicle by intraperitoneal injection once weekly for 14 weeks starting at 6.5 months of age and were subjected to the behavioral tasks (Figure 1). One and five groups of WT and Tg mice (seven mice per group), respectively, were treated as described in the Materials and Methods section (refer to Table 1).

The BW of each mouse was regularly monitored once a week, immediately after intraperitoneal administration, for the entire experimental period. There were no significant differences in average BW among experimental groups during the treatment period, indicating that neither the WV nor BV samples were toxic at the doses used in this study (Figure 2A).

A series of three behavioral tasks, including the passive avoidance test (PAT), Morris water maze test (MWM), and Y-maze test (YMT), were performed to examine the effect of WV administration on the learning and memory functions in 5xFAD mice (Figure 2B–D). The 5xFAD mice displayed cognitive and behavioral impairments in all tasks compared with the WT mice. In particular, the PAT, a fear-motivated associative learning and memory task, revealed that WV treatment at ≥250 μg/kg BW/day significantly rescued the memory deficits to the level of the groups treated with either BV at 25 μg/kg BW/day or donepezil at 5 mg/kg BW/day (Figure 2B). The MWM test showed that WV treatment at 250 μg/kg BW/day improved spatial learning and memory, indicating higher effectiveness than treatment with BV (25 μg/kg BW/day) or a high dose of WV (400 μg/kg BW/day) (Figure 2C). The YMT showed that WV treatment attenuated working memory deficits in 5xFAD mice (Figure 2D).

### 2.2. WV Treatment Ameliorates Histological Damage and Aβ Deposition in the Hippocampus

After behavioral studies, whole brains were removed from two mice per group, fixed in formalin solution, sliced, and stained with hematoxylin and eosin (H&E) or thioflavin S (Figure 3).

Brains from vehicle-treated 5xFAD Tg mice demonstrated substantial neuronal loss and disordered neuronal arrangement in the hippocampal CA1 and suprapyramidal blade of the dentate gyrus (Figure 3A), and more numerous thioflavin S-positive Aβ deposits compared to WT mice (Figure 3B). Notably, WV treatment at 250 μg/kg BW/day attenuated histological damage and decreased Aβ plaque formation in the hippocampal area. Consistent with these effects on plaque formation, the protein level of 99-amino acid C-terminal fragment of APP (C99) was significantly lower in hippocampal tissue homogenate of WV-treated Tg mice compared to vehicle-treated Tg mice (Figure 4A), while expression of the cleavage enzyme BACE1 was unaffected.

### 2.3. WV Treatment Decreases the Expression of Inflammation-Associated Markers in the Hippocampus and Cerebral Cortex

The WT treatment also significantly reduced the expression levels of cytoplasmic COX-2 in the hippocampus and nuclear NF-κB in the cerebral cortex of Tg mice compared to vehicle treatment (Figure 4A,B). Alternatively, there was only a modest reduction in COX-2 expression within the cerebral cortex following WV treatment. Nonetheless, these findings demonstrate that WV can suppress neuroinflammation in the hippocampal region of 5xFAD mice, consistent with effects on histopathology and memory impairments.

### 2.4. WV Treatment Attenuates Oxidative Stress in 5xFAD Mice

Oxidative stress was assessed by measuring levels of the lipid peroxidation product MDA and the DNA oxidation product 8-OHdG in the brain, liver, and plasma. We found that MDA levels were significantly elevated in the cerebral cortex of 5xFAD mice compared to WT mice, and that, however, treatment with 400 μg/kg BW/day WV significantly reduced marker expression (Figure 5A). More specifically, the MDA level in cerebral tissue homogenates from the vehicle-treated Tg mice was 7.9 nmoles/mg protein. However, the level in the Tg mice treated with WV at 400 μg/kg BW/day was 2.8 nmoles/mg protein.

Similarly, hepatic MDA level was reduced by 400 μg/kg BW/day WV to the level of WT mice (Figure 5B; 5.2 nmoles/mg protein for vehicle-treated WT mice, 6.6 nmoles/mg protein for vehicle-treated Tg mice, 4.4 nmoles/mg protein for DONE-treated Tg mice, and 3.4 nmoles/mg protein for 400 μg WV/kg BW/day-treated Tg mice).

Meanwhile, the plasma level of 8-OHdG in 5xFAD mice was also significantly reduced by WV treatment compared to vehicle treatment, although its levels in vehicle-treated 5xFAD Tg mice were highly variable and did not differ statistically from WT mice (Figure 5C). To be specific, the average plasma 8-OHdG level in the vehicle-treated Tg mice was 13.4 ng/mL. Treatment with WV at 250 μg/kg BW/day and 400 μg/kg BW/day substantially reduced the level to 6.0 and 6.1 ng/mL, respectively.

## 3. Discussion

Alzheimer’s disease (AD) is the most frequent cause of age-related dementia, accounting for an estimated 60–70% of all cases globally. The accumulation of Aβ accumulation and the subsequent deposition of hyperphosphorylated tau play a central role in the pathogenesis of AD. Aβ peptides ranging from 38 to 43 amino acids are generated through the cleavage of APP by BACEs, forming Aβ oligomers, polymers, and eventually insoluble Aβ plaques. Together with Aβ plaques, abnormal hyperphosphorylated tau tangles were hypothesized to be responsible for causing neuronal damage and subsequent cognitive impairments. 

Recently, it has been well acknowledged that neuroinflammation plays an important role in the pathogenesis of AD [21,22,23,24]. Consistently, this study demonstrated that anti-inflammatory activity is closely associated with the improvement of memory deficits in the 5xFAD mouse model expressing human APP and presenilin-1 harboring AD-linked mutations. We found that long-term treatment with WV reduced the levels of cytosolic COX-2 and nuclear NF-κB, which were relatively high in the 5xFAD mouse brain, and also improved cognitive functions which were impaired in the mouse model.

While BV has been widely used as an acupuncture agent for the treatment of rheumatoid arthritis, osteoarthritis, and AD in Korea, WV is not well studied for its clinical utility [25]. However, a few studies have examined the pharmacological effects of WV in the treatment of pain, inflammation, and neurodegenerative diseases [26]. Recently, we have reported that WV can inhibit LPS-induced microglial cell activation by suppressing NF-κB-mediated signaling pathways [7]. The findings from this study further demonstrated that acute treatment of WV can ameliorate cognitive impairments in LPS-treated WT mice, and long-term administration of WV improved the symptoms of AD, presumably by alleviating the inflammatory response and oxidative stress provoked in the brain of the 5xFAD Tg mouse.

Interestingly, the Aβ accumulation, as assessed by thioflavin S staining, was substantially suppressed in 5xFAD mice by WV administration, and this suppression could not be explained by reduced BACE1 activity. We speculate that Aβ is scavenged or degraded due to elevated expression of phase 2 detoxifying enzymes and/or promotion of the autophagy-lysosomal pathway in WV-treated mice. For instance, WV constituents may directly or indirectly interact with the Nrf2-Keap complex, activate Nrf2, and subsequently drive p62-mediated autophagy [27].

Meanwhile, the effective dose of WV for anti-inflammatory, antioxidant, and nootropic activity was approximately 10-fold higher than that of BV as assessed by mouse models. However, WV cytotoxicity was also at least 13-fold lower than BV according to our previous in vitro study [7]. Therefore, it is presumed that the bioactive ingredient(s) in WV are diluted approximately 10-fold compared to BV, and the side effects of WV are expected to be less severe than those of BV. More specifically, the advantages of WV over BV as a treatment/prophylactic agent for AD may include its low toxicity and presence of a different bioactive component profile from BV, suggesting that WV could be used as an alternative therapy for patients with AD who have developed a resistance to BV.

Although this study supports the memory-improving effect of WV in the AD mouse model, the active compounds responsible for the cognitive enhancement are still unclear. We attempted to purify the WV component(s) involved in the cognitive improvement using bioassay-guided fractionation and succeeded in identifying serotonin as a potential bioactive component. Consistently, serotonin found in the venom of *Vespa velutina nigrithorax* has been reported as a potent antioxidant [28].

However, it is generally believed that serotonin cannot cross the blood-brain barrier (BBB), and therefore, it is uncertain at present whether serotonin in WV is the bioactive substance responsible for improving memory function in mice that were intraperitoneally injected with WV. Previous studies have reported that serotonin could be converted into metabolite(s) that can penetrate the BBB, and the resultant metabolite(s) can probably exert their pharmacological effect. Another possibility is the conversion of serotonin to a BBB-permeable 5-hydroxy tryptophan or melatonin, which was reported to improve AD symptoms, in tissue or blood before being transported to the brain [29,30].

In conclusion, the findings from this study demonstrated that WV can attenuate learning and memory impairments in a mouse model recapitulating the core features of AD, possibly by suppressing NF-κB-mediated neuroinflammation and oxidative stress. However, further studies are needed to identify the component(s) in WV that are responsible for the memory-enhancing effect in vivo and elucidate their mode of action.

## 4. Materials and Methods

### 4.1. Preparation of WV and BV

*Vespa velutina nigrithorax* colonies (10 nests, weighing 3–4 kg/nest) were collected in South Korea during August and October of 2020 and stored at −80 °C until venom extraction. The WV sample used in this study was prepared as previously described [7]. Briefly, the venom sac was manually removed from each wasp and collected in Spin-X 0.45-μm cellulose acetate filter tubes (Corning Inc., Salt Lake City, UT, USA). After a brief centrifugation, the filtrate was freeze-dried and dissolved in an appropriate solvent for further experiments. BV powder containing melittin (40–50%), apamin (2–3%), phospholipase A2 (10–13%), histamine (0.5–2%), dopamine (0.2–1%), MCD-peptide 401 (2–3%), and others was purchased from Chung Jin Biotech (Ansan, Republic of Korea).

### 4.2. Experimental Animals

Male hemizygous 5xFAD Tg mice on the 57BL/6J genetic background (B6.Cg-Tg(APPSwFlLon,PSEN1*M146L*L286V)6799Vas/Mmjax; MMRRC Strain #034848-JAX) were obtained from Jackson Laboratory (Bar Harbor, ME, USA) [19] and bred with congenic wild-type (WT) C57BL/6J females (Daehan Biolink, Eumsung, Republic of Korea). After weaning, offspring were housed together until WV harvesting. Male C57BL/6J WT mice from Daehan Biolink were used as controls. All mice were housed in standard conditions (ambient temperature, 20–25 °C; relative humidity, 45% ± 5%; 12 h/12 h light/dark cycle) and fed a standard rodent diet (Daehan BioLink, Seoul, Republic of Korea).

### 4.3. Genotyping

Genomic DNA was extracted from the tail for genotyping. Briefly, the tail sample was incubated overnight in a solution containing TRI Reagent (Sigma-Aldrich, St. Louis, MO, USA) and Proteinase K (RBC Bioscience, New Taipei City, Taiwan), followed by addition of 0.2 mL chloroform, vortexing, and centrifugation at 12,000× *g* for 5 min. The aqueous layer was retained and mixed with 0.3 mL of 100% ethanol, followed by vortexing, and centrifugation at 5000× *g* for 5 min. The pelleted DNA was collected, washed with 75% ethanol, dried, and dissolved in DNAase-free, purified water. Genotyping was performed using the following primers: (1) 5′-ACC CCC ATG TCA GAG TTC CT-3′ (Chr3, common); (2) 5′-TAT ACA ACC TTG GGG GAT GG-3′ (Chr3, WT reverse); and (3) 5′-CGG GCC TCT TCG CTA TTA C-3′ (vector, mutant reverse) as per the Jackson Laboratory’s protocols. It was expected that heterozygous and wild-type mice would yield 129-bp and 216-bp amplicons, respectively.

### 4.4. Experimental Design

All animal study protocols were conducted according to the guidelines of the Institutional Animal Care and Use Committee of Kyungpook National University (Daegu, Republic of Korea) (approval number: KNU 2021-0078). Thirty-five 5xFAD Tg male mice were randomly assigned equally into five groups of seven, and seven C57BL/6J WT male mice were included as a control group (refer to Table 1). Groups were treated as follows: (1) WT mice treated with vehicle only, (2) Tg mice treated with vehicle only, (3) Tg mice treated with donepezil (DONE) at 5 mg/kg body weight (BW)/day, (4) Tg mice treated with BV at 25 μg/kg BW/day, (5) Tg mice treated with WV at 250 μg/kg BW/day, and (6) Tg mice treated with WV at 400 μg/kg BW/day. The vehicle consisted of 2% (*v*/*v*) Tween^®^ 80 solution dissolved in normal saline (Sigma-Aldrich, Burlington, MA, USA).

### 4.5. Behavioral Tests

The passive avoidance test (PAT) for associative memory, the Y-maze test (YMT) for working memory, and the Morris water maze test (MWM) for spatial learning and long-term memory were conducted following treatments according to procedures described previously [31,32,33,34].

The PAT is a tool for evaluating fear-motivated learning and memory. The apparatus (Gemini Avoidance System, San Diego, CA, USA) was composed of two chambers (dark and bright) and a guillotine door in between the two chambers. On the first day of the test, mice treated as scheduled (refer to Figure 1) were placed in the bright chamber with the door open and allowed to adapt the equipment for 1 min. The next day for the training trial, each mouse was placed in the bright chamber with the door closed. Ten seconds later, the door was opened. At the time when the mouse entered the dark chamber, the door was closed, and the latency time spent in the bright chamber was recorded (‘training’). Then, an electrical foot shock (0.5 mA) was delivered for 3 sec through the stainless-steel rods in the dark chamber. The day after, each mouse was again placed in the bright chamber with the door closed. Ten seconds later, the door was opened. At the time when the mouse moved into the dark chamber, the latency time staying in the bright chamber was recorded (‘test’). If a mouse did not enter the dark chamber within 300 sec, the mouse was removed and assigned a latency score of 300 sec. Short latency times represent poor learning and memory abilities.

The YMT is a tool for studying working memory by measuring the spontaneous alternation of mice among multiple arms of the Y-shaped maze. A single 5-min test was performed in which each mouse was placed in the A arm of the maze. For each mouse, the total number of arm entries and the order of entries during the Y-maze trials were recorded. Spontaneous alternations were defined as consecutive triplets of different arm choices. The percentage of alternations was calculated according to the following formula: % alternation = [(number of alternations)/(total arm entries − 2)] × 100.

The MWM was performed in a circular swimming pool (90 cm in diameter and 45 cm in height) with a featureless inner surface. The pool was filled with water (22  ±  2 °C) to a depth of 28 cm. For the first day of the MWM, mice were allowed to swim freely for 1 min. A black platform (6 cm in diameter and 29 cm in height) was then placed in one of the pool quadrants. The water was made opaque with nontoxic poster paint. For the next three consecutive days, each mouse was given three trials per session per day to locate the platform in place for 1 min. If the mouse did not find the platform within the given time, it was gently guided to a place on the platform for 10 sec. On the fifth day, the platform was submerged 1 cm below the opaque water level in the same quadrant. The swimming time for each mouse to locate the platform was recorded, analyzed, and graphed.

### 4.6. Western Blotting and Analysis

Dissected brain tissues were homogenated manually and fractionated into cytoplasmic and nuclear components using a NE-PER^®^ nuclear and cytoplasmic extraction reagent (Thermo Fisher Scientific, Waltham, MA, USA) as per the manufacturer’s instructions. The extracted proteins were quantified by performing the Bradford assay [35]. An equal amount of the proteins from the lysates were loaded onto a 10% polyacrylamide gel. The electrophoretically separated proteins were then transferred onto a polyvinylidene fluoride (PVDF) membrane (Immobilon^®^–P; Millipore, Burlington, MA, USA). Proteins were detected using primary antibodies: mouse anti-C99 (Millipore), mouse anti-BACE1 (Santa Cruz Biotechnology, Dallas, TX, USA), rabbit anti-COX-2 (Cell Signaling Technology, Danvers, MA, USA), rabbit anti-NF-κB (Cell Signaling Technology), mouse anti-β-actin (Santa Cruz Biotechnology), or rabbit anti-lamin B (Cell Signaling Technology), and appropriate secondary antibodies that were labeled with horseradish peroxidase. The antibody-bound proteins were visualized using the SuperSignal™ West Pico Chemiluminescent Substrate (Pierce, Cheshire, UK) and LAS4000 Mini (GE Healthcare Life Sciences, Little Chalfont, UK). The protein bands were densitometrically analyzed using the Image Studio Lite version 5.2 software (Ll-COR Biotechnology, Lincoln, NE, USA).

### 4.7. Measurement of Oxidative Stress Biomarkers

Malondialdehyde (MDA) is a byproduct of lipid peroxidation caused by oxidative stress in tissues. The liver and cerebral cortex were dissected from the sacrificed animals and homogenized in 0.1 M phosphate buffer (pH 7.4). After centrifugation at 10,000× *g* for 30 min at 4 °C, the supernatant of the homogenates was used for determining the MDA level. The MDA level was quantified by performing the thiobarbituric acid-reactive substances assay using a commercially available kit (Cat. # ALX-850-087; Enzo Life Sciences, Inc., Plymouth Meeting, PA, USA) by following the manufacturer’s instructions. The absorbance of the resulting reaction mixture was detected at 532 nm. The values were normalized to the total amount of protein [35,36].

8-hydroxy-2′-deoxyguanosine (8-OHdG) is a biomarker for detecting DNA damage caused by oxidative stress. The plasma was collected from the whole blood of the mice after finishing the experiment and used for the determination of the 8-OHdG level using a commercially available kit (Cat. # ADI-EKS-350; Enzo Life Sciences, Inc.). The absorbance of the resulting reaction mixture was measured at 450 nm.

### 4.8. Histological Analysis

The brain tissues were dissected from the mice. Brain tissues were isolated, fixed in formalin, embedded in paraffin blocks, and sectioned at 5-µm thickness using a microtome (RM-2125 RT; Leica, Nussloch, Germany). These slices were placed on microscope slides (Marienfeld, Lauda-Königshofen, Germany) and were then stained using hematoxylin and eosin (H&E) dyes (Sigma-Aldrich). The slices on the microscope slides were first deparaffinized in xylene, rehydrated in gradient alcohol, and stained with H&E dyes. After rinsing in running water, the tissue slices were dehydrated by sequentially soaking them in alcohol and xylene. The stained slices were then mounted in malinol medium (Muto pure chemical, Tokyo, Japan) and xylene and examined under an optical microscope.

### 4.9. Thioflavin S Staining

Tissue sections prepared as described in Section 4.8 were deparaffinized, rehydrated, and immersed in a solution of 0.002% (*w*/*v*) thioflavin S (Sigma-Aldrich) and 50% ethanol. After sequential rinsing with 50% ethanol and PBS, nuclei were counterstained with 4′,6-diamidino-2-phenylindole dihydrochloride (DAPI). The sections were mounted using a commercially available mounting medium (Dako Denmark A/S, Glostrup, Denmark). The thioflavin-S-labeled Aβ plaques and DAPI-stained nuclei were observed under a fluorescence microscope.

### 4.10. Statistical Analysis

All results are expressed as the mean ± standard deviation (SD). Treatment group means were compared by one-way analysis of variance followed by Duncan’s multiple range tests using SPSS Statistics 22 software (SPSS Inc., Chicago, IL, USA). A *p*-value less than 0.05 was considered statistically significant for all tests. In the figures, statistically significant differences are marked using different alphabetical letters.

## Figures and Tables

**Figure 1 toxins-15-00203-f001:**
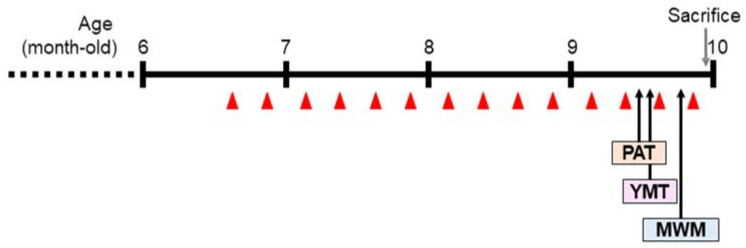
Experimental schedule. Wasp venom (WV) or bee venom (BV) was administered by intraperitoneal injection to 6.5-month-old 5xFAD mice once per week for a total of 14 weeks. PAT, passive avoidance test; YMT, Y-maze test; MWM, Morris water maze test. The red triangle denotes the points at which samples were injected.

**Figure 2 toxins-15-00203-f002:**
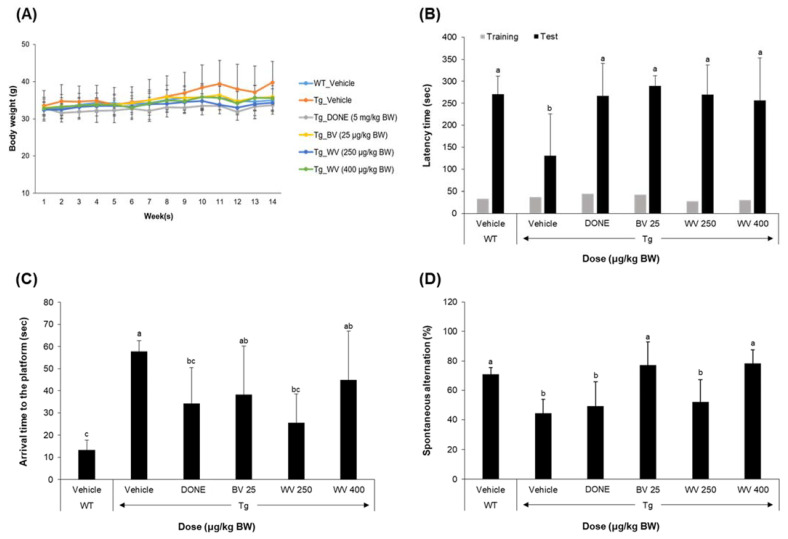
Wasp venom improved associative, working, and spatial memory in 5xFAD Tg mice. Seven WT mice (C57BL/6J, male) and 35 Tg mice were allocated to six experimental groups (seven mice per group) as follows: (1) WT_Vehicle, WT mice treated only with vehicles; (2) Tg_Vehicle, 5xFAD mice treated only with vehicles; (3) Tg_DONE, 5xFAD mice treated with 5 mg/kg BW/day DONE; (4) Tg_BV 25, 5xFAD mice treated with 25 μg/kg BW/day BV; (5) Tg_WV 250, 5xFAD mice treated with 250 μg/kg BW/day WV; and (6) Tg_WV 400, 5xFAD mice treated with 400 μg/kg BW/day WV. BW, body weight; DONE, donepezil; BV, bee venom; WV, wasp venom. (**A**) Body weight changes over the experimental period did not differ among groups. (**B**) Associative learning and memory were assessed by the PAT. In Tg mice, all treatments increased approach latency to the level of WT controls. (**C**) Spatial learning and memory were assessed by the MWM. (**D**) Working memory was assessed by the YMT. Values are expressed as mean ± SD (*n* = 7). Values marked with different letters differ statistically significantly from one another at *p* < 0.05.

**Figure 3 toxins-15-00203-f003:**
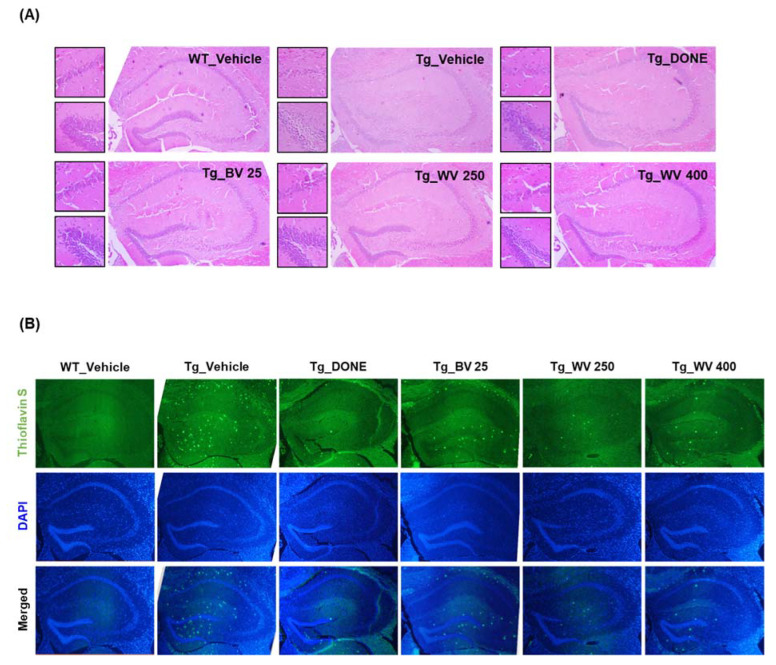
Wasp venom ameliorated histological damage and Aβ deposition in the hippocampal area of 5xFAD mice. (**A**) Representative images of brain slices stained with H&E. Images on the right-side of the panel were taken at 40× magnification. Small images on the left-side were acquired at 100× magnification, focusing on the hippocampal CA1 region (**upper**) and the dentate gyrus (**lower**). (**B**) Representative images of brain slices stained with thioflavin S and DAPI (40× magnification). WT, wild type; Tg, transgenic; DONE, donepezil; BV, bee venom; WV, wasp venom; DAPI, 4′,6-diamidino-2-phenylindole dihydrochloride.

**Figure 4 toxins-15-00203-f004:**
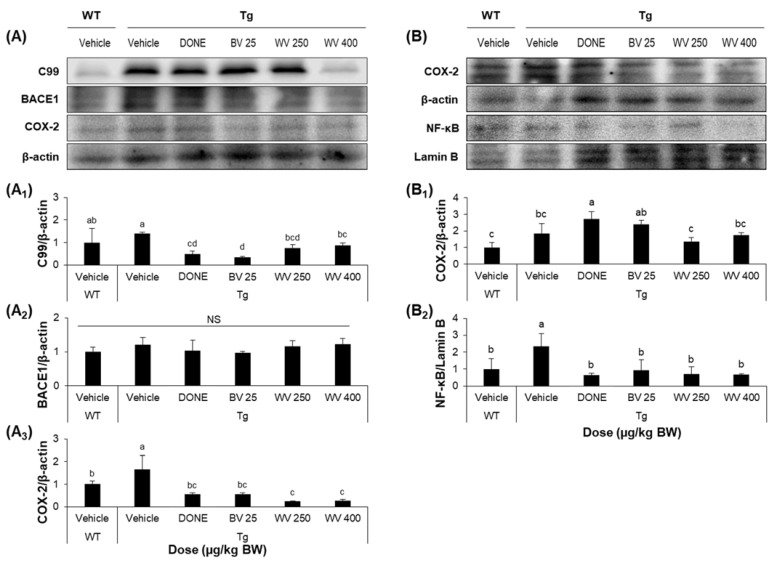
Wasp venom reduced the expression levels of inflammatory markers in the hippocampus and cerebral cortex of 5xFAD mice. (**A**) Representative Western blots images of cytoplasmic C99, BACE1, and COX-2 protein levels in hippocampal tissue homogenates. (**A_1_**–**A_3_**) Quantitative analysis of the levels of C99 (**A_1_**), BACE1 (**A_2_**), and COX-2 (**A_3_**) relative to β-actin. (**B**) Representative Western blots images of cytoplasmic COX-2 and nuclear NF-κB protein levels in cerebrocortical tissue homogenates. (**B_1_**,**B_2_**) Quantitative analysis of the levels of COX-2 (**B_1_**) and NF-κB (**B_2_**) relative to β-actin and Lamin B, respectively. Refer to Appendix A for protein bands from all tissue homogenates. Values are expressed as mean ± SD (*n* = 5). Values marked with different letters differ statistically significantly from one another at *p* < 0.05.

**Figure 5 toxins-15-00203-f005:**
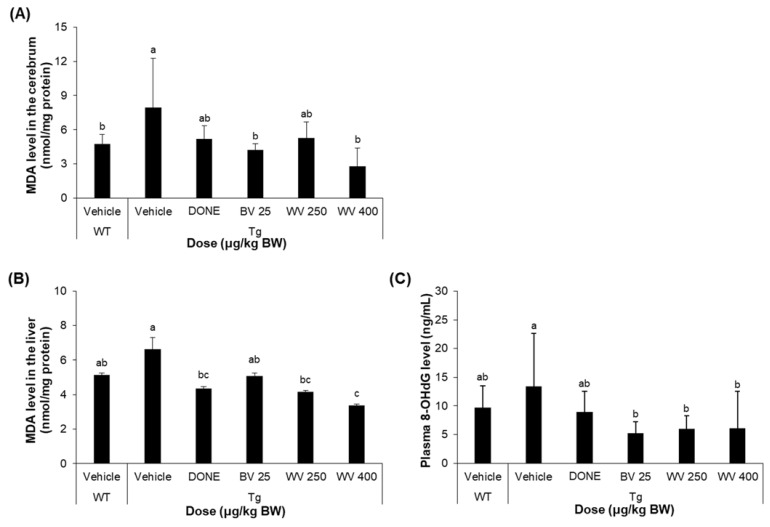
Wasp venom decreased oxidative stress in 5xFAD mice. (**A**) MDA level in the cerebral cortex (*n* = 5). (**B**) MDA level in the liver (*n* = 5). (**C**) 8-OHdG level in the plasma (*n* = 7). Values are expressed as mean ± SD. Values marked with different letters differ statistically significantly from one another at *p* < 0.05.

**Table 1 toxins-15-00203-t001:** Experimental groups used in the animal study.

Group	Experimental Groups (*n* = 7 per Group)					
	1	2	3	4	5	6
Mouse type	WT (C57BL/6J)	5xFAD Tg				
Sample treatment (*i.p.*)	Vehicle	Vehicle	DONE	BV	WV	
	-	-	5 mg/kg BW/day	25 μg/kg BW/day	250 μg/kg BW/day	400 μg/kg BW/day

WT, wild-type; Tg, transgenic; *i.p.*, intraperitoneal injection; DONE, donepezil; BV, bee venom; WV, wasp venom.

## Data Availability

The authors declare that all data supporting the findings of this study are available in the article and can be provided by the corresponding author on reasonable request.

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
