# Peer review of "Long-Term Administration of Vespa velutina nigrithorax Venom Ameliorates Alzheimer’s Phenotypes in 5xFAD Transgenic Mice"

_toxins, 2023, doi:10.3390/toxins15030203_

Round 1
Reviewer 1 Report
“Long-term Administration of Wasp Venom Ameliorates Alzheimer’s Phenotypes in 5xFAD Transgenic Mice”
In the above paper, Vespa velutina venom is administered intraperitoneal (two concentrations used: at 250 and 400 μg/kg body weight (BW); once weekly (n=14 weeks) to wild type mice and 5xFAD transgenic mice. Other groups: Wild mice treated with “vehicle”, and 5xFAD transgenic mice treated with donepezil (DONE, 5mg/kg/BW/day) and 5xFAD transgenic treated with bee venom (BV, 25 micrograms/kg/BW/day). 3 behavioral tests [the passive avoidance test (PAT) for associative memory, Y-maze test (YMT) for working memory, and Morris water maze test (MWM) for spatial learning and long-term memory] were used in the different treatment groups. Furthermore , oxidative stress biomarkers and histological analysis were carried out in the above groups. The MWM test showed that vespa velutina venom treatment at 250 μg/kg BW/day improved the spatial learning and memory. Long-term treatment with vespa velutina venom reduced the levels of cytosolic COX-2 and nuclear NF-κB, which were relatively high in 5xFAD mouse brain. The YMT showed that vespa velutina venom treatment attenuated working memory deficits in 5xFAD mice.
I find the work carried out very interesting. I have a number of comments to the authors, which could perhaps improve the manuscript and its clarity.
Do not the authors believe that the wild mice, in addition to being treated with the vehicle, should have been formed and studied groups treated with bee venom and also with donezepil, as is done with transgenic mice?
I suggest that the scientific name of the species be introduced in the title: Vespa velutina nigrithorax
There is room to improve the introduction. From my point of view, supplementary material should not be included in this section (Figures s1 to s33). Lines 35 to 39, must be rewritten. Lines 66-67 relative to supplementary figures should be removed.
It might be appropriate to refer specifically to hornet venom, and not to wasps in general. And briefly refer to the different works on hornet venoms of different species, their therapeutic potentials or molecules found with their physiological properties. It would also enrich the introduction to point out the species as invasive (both in South Korea and in Europe) and its allergological and epidemiological characteristics. Consider introducing the following references related with the species and the venom:
-Vidal, C. The Asian wasp Vespa velutina nigrithorax: Entomological and allergological characteristics. Clin Exp Allergy 2022, 52 (4), 489-498. DOI: 10.1111/cea.14063.
-Feás, X. Human Fatalities Caused by Hornet, Wasp and Bee Stings in Spain: Epidemiology at State and Sub-State Level from 1999 to 2018. Biology (Basel) 2021, 10 (2). DOI: 10.3390/biology10020073.
-Feás, X.; Vidal, C.; Vázquez-Tato, M. P.; Seijas, J. A. Asian Hornet, Vespa velutina Lepeletier 1836 (Hym.: Vespidae), Venom Obtention Based on an Electric Stimulation Protocol. Molecules 2021, 27 (1). DOI: 10.3390/molecules27010138.
-Choi, M. B.; Kim, T. G.; Kwon, O. Recent Trends in Wasp Nest Removal and Hymenoptera Stings in South Korea. J Med Entomol 2019, 56 (1), 254-260. DOI: 10.1093/jme/tjy144.
The results must be presented in the main text, not all of them logically, but a few, the most important, … using their units (in seconds, mmol/mg protein, ng/mL ..)
In Figure 4, please provide in the axys or in the figure caption the units of: cytoplasmic C99, BACE1, COX-2 levels in hippocampal tissue homogenates; cytoplasmic COX-2 and nuclear NF-κB protein levels in cerebrocortical tissue homogenates
Line 209-211
The authors comment “We attempted to purify the WV component(s) involved in the cognitive improvement using bioassay-guided fractionation and succeeded in identifying serotonin as a potential bioactive component”.
I think they should present the methods used, as well as the results obtained to conclude the identification of serotonin and its responsibility as a potential bioactive component in cognitive improvement. In this sense, it is good to cite and discuss the previous results of:
LE, T. N.; Da Silva, D.; Colas, C.; Darrouzet, E.; Baril, P.; Leseurre, L.; Maunit, B. Asian hornet Vespa velutina nigrithorax venom: Evaluation and identification of the bioactive compound responsible for human keratinocyte protection against oxidative stress. Toxicon 2020, 176, 1-9. DOI: 10.1016/j.toxicon.2020.01.001.
It is necessary to specify a little more how the insects were obtained. How many nests, where... In this sense, a variation has been seen in certain components of the velutin venom depending on the harvest season.
See:
Le, T. N.; da Silva, D.; Colas, C.; Darrouzet, E., Baril, P., Leseurre, L., Maunit, B. Development of an LC-MS multivariate nontargeted methodology for differential analysis of the peptide profile of Asian hornet venom (Vespa velutina nigrithorax): application to the investigation of the impact of collection period variation. Anal Bioanal Chem 2020, 412 (6), 1419-1430. DOI: 10.1007/s00216-019-02372-2. )
I also think that it should be presented, since I understand that it is a standard obtained from a company, characteristics of the bee venom used (compounds and concentrations). The concentrations of bee venoms used, … why they were chosen (25 micrograms/kg/BW/da ) and for velutina venom (250 and 400 μg/kg body weight (BW))?
How many milliliters of vehicle were injected in the mouse each time?
Supplementary mat (lines 362-393). It is not neccessary to provide full figure captions an this point.
Author Response
In the above paper, Vespa velutina venom is administered intraperitoneal (two concentrations used: at 250 and 400 μg/kg body weight (BW); once weekly (n=14 weeks) to wild type mice and 5xFAD transgenic mice. Other groups: Wild mice treated with “vehicle”, and 5xFAD transgenic mice treated with donepezil (DONE, 5mg/kg/BW/day) and 5xFAD transgenic treated with bee venom (BV, 25 micrograms/kg/BW/day). 3 behavioral tests [the passive avoidance test (PAT) for associative memory, Y-maze test (YMT) for working memory, and Morris water maze test (MWM) for spatial learning and long-term memory] were used in the different treatment groups. Furthermore, oxidative stress biomarkers and histological analysis were carried out in the above groups. The MWM test showed that vespa velutina venom treatment at 250 μg/kg BW/day improved the spatial learning and memory. Long-term treatment with vespa velutina venom reduced the levels of cytosolic COX-2 and nuclear NF-κB, which were relatively high in 5xFAD mouse brain. The YMT showed that vespa velutina venom treatment attenuated working memory deficits in 5xFAD mice.
I find the work carried out very interesting. I have a number of comments to the authors, which could perhaps improve the manuscript and its clarity.
Response: We would like to express our deepest gratitude for the time and effort you have dedicated to reviewing our manuscript. Your thoughtful comments and suggestions have been instrumental in improving the quality of our work. Your valuable feedback and constructive criticism have helped us to refine our research and presentation.
Do not the authors believe that the wild mice, in addition to being treated with the vehicle, should have been formed and studied groups treated with bee venom and also with donezepil, as is done with transgenic mice?
Response: Thank you very much for the appropriate comments.
The reason that the groups of WT mice treated with BV or WV or donepezil were not included in this study
- The hypothesis to test in the present study was that the administration of wasp venom (WV) isolated from Vespa velutina nigrithorax may improve AD phenotypes through attenuating neuroinflammation in vivo.
- As we have briefly addressed in the Introduction section, it was rationalized by two primary observations: (1) the WV suppressed the activation of BV-2 microglial cells in response to LPS treatment, and (2) intraperitoneal administration of WV to LPS-treated wild-type C57BL/6J mice improved neuroinflammation-induced memory deficits.
- Thus, we believe that WV administration can dampen the neuroinflammatory response in the brain which is a major contributor to the pathogenesis of neurodegenerative disorders including AD. In this sense, we have examined he effectiveness of WV administration in the mice which exhibited learning and memory impairments.
Determination of injection doses of BV and WV
- As the supplementary information shows, the beneficial effect of BV and WV was examined in LPS-treated mice.
- The doses of those venoms were determined on the basis of our in vitro toxicity test (refer to a published paper, DOI: 10.3390/insects12040297) and preliminary in vivo toxicity test.
- For the in vivo toxicity test, we performed in WT C57BL/6J mice with various doses of BV and WV only from 5 to 500 μg/kg BW/day which were injected every day for two weeks.
- Therefore, the doses of BV and WV used in 5xFAD Tg mice were determined at which the venoms caused no apparent abnormalities in mice including body weight loss.
I suggest that the scientific name of the species be introduced in the title: Vespa velutina nigrithorax
Response: The title was changed to “Long-term Administration of Vespa velutina nigrithorax Venom Ameliorates Alzheimer’s Phenotypes in 5xFAD Transgenic Mice” as suggested.
There is room to improve the introduction. From my point of view, supplementary material should not be included in this section (Figures s1 to s33). Lines 35 to 39, must be rewritten. Lines 66-67 relative to supplementary figures should be removed.
Response: Lines 35-39 in the Abstract section were revised. We attempted to summarize the results described in main text exclusively but not in supplements.
The supplementary data, Figures S1-S3, in the Introduction section
- The data demonstrates that WV administration can improve inflammation-induced memory injury in LPS-treated wild-type C57BL/6J mice. Those have not been published elsewhere.
- In fact, this data prompted us to undertake the present study investigating the beneficial effect of long-term administration of WV in an AD mouse model, since neuroinflammation (the inflammatory process in the brain that leads to glial cell activation and further to neuronal degeneration) is widely recognized as a prominent feature of AD.
- Thus, we believe that it should be addressed before getting into the main text where we profoundly focused on the results from the transgenic mice.
It might be appropriate to refer specifically to hornet venom, and not to wasps in general.
Response: We respect the reviewer’s suggestion regarding the word of hornet venom instead of wasp venom. When searched in Google Scholar, hornet venom and wasp venom were used in almost similar frequency. To avoid any ambiguity, we would like to retain the term of ‘wasp venom’ for this manuscript.
And briefly refer to the different works on hornet venoms of different species, their therapeutic potentials or molecules found with their physiological properties. It would also enrich the introduction to point out the species as invasive (both in South Korea and in Europe) and its allergological and epidemiological characteristics. Consider introducing the following references related with the species and the venom:
-Vidal, C. The Asian wasp Vespa velutina nigrithorax: Entomological and allergological characteristics. Clin Exp Allergy 2022, 52 (4), 489-498. DOI: 10.1111/cea.14063.
-Feás, X. Human Fatalities Caused by Hornet, Wasp and Bee Stings in Spain: Epidemiology at State and Sub-State Level from 1999 to 2018. Biology (Basel) 2021, 10 (2). DOI: 10.3390/biology10020073.
-Feás, X.; Vidal, C.; Vázquez-Tato, M. P.; Seijas, J. A. Asian Hornet, Vespa velutina Lepeletier 1836 (Hym.: Vespidae), Venom Obtention Based on an Electric Stimulation Protocol. Molecules 2021, 27 (1). DOI: 10.3390/molecules27010138.
-Choi, M. B.; Kim, T. G.; Kwon, O. Recent Trends in Wasp Nest Removal and Hymenoptera Stings in South Korea. J Med Entomol 2019, 56 (1), 254-260. DOI: 10.1093/jme/tjy144.
Response: Thank you very much for introducing several good articles.
As the reviewer suggested, we have elaborated the Introduction section with those references cited and additional sentences inserted to describe the different actions of wasp venom components of different species including mastoparans.
The results must be presented in the main text, not all of them logically, but a few, the most important, … using their units (in seconds, mmol/mg protein, ng/mL ..)
Response: As suggested, the section 2.4 was moderately revised.
We have prioritized and logically organized our data while also presenting the significance and critical points for each data set (related to venom doses) in a concise manner through figures, in order to facilitate reader comprehension.
- Results in Figure 2 are the data from behavioral tasks. All the data were summarized in the graphs to show the results in an intuitive manner. If needed, we can provide the table including the information on the latency time, arrival time to the platform, and spontaneous alternation in detail as an additional supplementary material.
- Regarding Figure 4 from Western blot analyses, the data are consisted of relative values. It means that the expression level of a specific protein was normalized to the constitutively expressing protein (such as beta-actin for the cytoplasmic fraction and lamin B for the nuclear fraction) and averaged from n = 5 samples.
- Figure 5 contains the results from the measurement of oxidative stress markers, MDA and 8-OHdG. The examinations were performed based on ELISA using commercially available kits. The resulting values were therefore obtained by extrapolating the experimental values to the standard curve. It means that the resulting values can differ from each brand of the assay kit and each laboratory.
- Therefore, we focused on whether the values are statistically different or not among the experimental groups and compared to the control condition, rather than expounding on the resulting data with the units.
In Figure 4, please provide in the axys or in the figure caption the units of: cytoplasmic C99, BACE1, COX-2 levels in hippocampal tissue homogenates; cytoplasmic COX-2 and nuclear NF-κB protein levels in cerebrocortical tissue homogenates
Response: Figure 4 caption was revised accordingly.
Line 209-211
The authors comment “We attempted to purify the WV component(s) involved in the cognitive improvement using bioassay-guided fractionation and succeeded in identifying serotonin as a potential bioactive component”.
I think they should present the methods used, as well as the results obtained to conclude the identification of serotonin and its responsibility as a potential bioactive component in cognitive improvement. In this sense, it is good to cite and discuss the previous results of:
LE, T. N.; Da Silva, D.; Colas, C.; Darrouzet, E.; Baril, P.; Leseurre, L.; Maunit, B. Asian hornet Vespa velutina nigrithorax venom: Evaluation and identification of the bioactive compound responsible for human keratinocyte protection against oxidative stress. Toxicon 2020, 176, 1-9. DOI: 10.1016/j.toxicon.2020.01.001.
Response: Another set of experiment has been conducted using serotonin isolated from the wasp venom and the details will be published as a different paper soon.
Instead the article that the reviewers recommended was briefly discussed in main text (Lines 266-267).
It is necessary to specify a little more how the insects were obtained. How many nests, where... In this sense, a variation has been seen in certain components of the velutin venom depending on the harvest season. See:
Le, T. N.; da Silva, D.; Colas, C.; Darrouzet, E., Baril, P., Leseurre, L., Maunit, B. Development of an LC-MS multivariate nontargeted methodology for differential analysis of the peptide profile of Asian hornet venom (Vespa velutina nigrithorax): application to the investigation of the impact of collection period variation. Anal Bioanal Chem 2020, 412 (6), 1419-1430. DOI: 10.1007/s00216-019-02372-2. )
Response: The details of wasp venom sample collection were described in the Materials and Methods section (4.1. Preparation of WV and BV).
- Vespa velutina nigrithorax colonies were collected in South Korea during August and October of 2020. More specifically, 10 of wasp nests were collected from different local areas of South Korea. The wasp nests were stored at −80°C until venom extraction.
- Adult female Vespa velutina nigrithorax wasps were identified by Dr. Moon-Bo Choi and selected for venom extraction. The extracted venoms were pooled prior to use as a sample for the study.
- Thus, the paragraph was revised accordingly.
I also think that it should be presented, since I understand that it is a standard obtained from a company, characteristics of the bee venom used (compounds and concentrations).
Response: We are thankful for the comment and newly described the compositions of BV which was used in the study in the M&M section 4.1. as follows; “BV powder containing melittin (40~50%), apamin (2-3%), phospholipase A2 (10-13%), histamine (0.5-2%), dopamine (0.2-1%), and MCD-peptide 401 (2-3%) and others was purchased from Chung Jin Biotech (Ansan, South Korea).”
The concentrations of bee venoms used, … why they were chosen (25 micrograms/kg/BW/da ) and for velutina venom (250 and 400 μg/kg body weight (BW))?
How many milliliters of vehicle were injected in the mouse each time?
Response: The concentrations of BV and WV were determined based on our previous in vitro and in vivo studies.
- We found that WV was at least 13-fold less cytotoxic than BV while WV was approximately 20-fold weaker than BV in most inflammatory parameters in vitro study (DOI: 10.3390/insects12040297).
- More importantly, the results from the in vivo study using LPS-treated WT mice (as shown in Supplementary Figures S1-S3) demonstrated that the memory-improving effect of WV at a dose of 50 μg/kg BW was comparable to that of BV at 5 μg/kg BW.
- In addition, we found that WV at a dose of 500 μg/kg BW increased the proinflammatory cytokines, TNFα and IL-1β.
- Taken all together, the doses of WV and BV were determined at which those venoms caused no apparent abnormalities in mice including body weight loss but exhibited a biologically beneficial effect.
Supplementary mat (lines 362-393). It is not neccessary to provide full figure captions an this point.
Response: We understand the reviewer’s point. However, we would like not to remove the Supplementary materials from this manuscript as addressed on their significance.
- Initially, we conducted a study on wild type C57BL/6J mice to investigate the effect of WV on lipopolysaccharide-induced inflammation and cognitive deficits. The findings from this study are presented in the supplementary material. Following this, we performed a similar study on 5xFAD mice and the results were presented in the main text of this manuscript. The supplementary material provides the background and validity for the current study, which was conducted using an AD mouse model (5xFAD mice). The results from both mouse models consistently demonstrate that wasp venom has a protective effect on AD-like phenotypes.
In addition, the section containing the captions for the supplementary materials is part of the journal format.
Reviewer 2 Report
In present work the Authors suggest that wasp venom obtained from Vespa velutina alleviates Alzheimer’s disease symptoms and delays its progression.
The subject of this study is noteworthy. The manuscript is well written. Each part of the manuscript is described in detail, the figures are accurate.
Future research in this direction is certainly needed.
Author Response
Thank you very much for encouraging comments.
Reviewer 3 Report
Abstract
Line 17-18 -> "Serotonin found in WV is may partially contribute in the protective effect exerted on the progression of AD." -> This phrase should be ommited from abstract -> It is a clear assumption. Authors do not show any result or characterization for serotonin as the WV effective component responsible for any improvement of AD symptoms. It is just an hypothesis as discussed in lines 208-219.
Line 18-20 -> "...WV may alleviate AD symptoms and delay its progression." -> review this sentence -> How can you write that WV delays AD progression as a conclusion? What is AD progression? This is not described at Introduction. Also, presented results and discussion are not clearly allowing such conclusion.
Key Contributions
Line 27-28 -> "...therapeutic potential for AD whose treatment currently exists." -> review -> Shouldn´t it be "currently does not exist"?
Discussion
Line 176 -> "Previous studies, this study..." -> review and correct;
Line 188 -> "While Although BV..." -> correct;
Line 192 -> "...the Aβ accumulation, Accumulation..." -> correct;
Line 223 -> "components(s)" -> correct;
Author Response
Response: We would like to express our deepest gratitude to the reviewer for taking a precious time and making the constructive comments.
Abstract
Line 17-18 -> "Serotonin found in WV is may partially contribute in the protective effect exerted on the progression of AD." -> This phrase should be ommited from abstract -> It is a clear assumption. Authors do not show any result or characterization for serotonin as the WV effective component responsible for any improvement of AD symptoms. It is just an hypothesis as discussed in lines 208-219.
Response: Thank you very much for the excellent comments. The sentence was removed from the Abstract section as suggested.
In terms of the effectiveness of serotonin, we are currently performing a set of study using 5xFAD mice. Hopefully, we can submit the manuscript related to the topic sooner than later.
Line 18-20 -> "...WV may alleviate AD symptoms and delay its progression." -> review this sentence -> How can you write that WV delays AD progression as a conclusion? What is AD progression? This is not described at Introduction. Also, presented results and discussion are not clearly allowing such conclusion.
Response: We agree to the referee’s point that the resulting data from this study did not address the progression of AD and that we did not diagnose AD according to the NIA-AA criteria defining the AD phases (preclinical AD, MCI, and dementia) in mice because of a practical difficulty.
Instead, our results demonstrated that WV treatment improved learning and memory impairments, reduced hippocampal injury and Aβ accumulation, and decreased oxidative stress markers in the brain, liver, and blood). Thus, the conclusion was revised to “these findings suggested that a long-term administration of WV may alleviate AD-related symptoms and pathological phenotypes.”
Key Contributions
Line 27-28 -> "...therapeutic potential for AD whose treatment currently exists." -> review -> Shouldn´t it be "currently does not exist"?
Response: Thank you very much for the comments on our typos.
It was revised as follows; “The observed beneficial effect of wasp venom on AD poses as an alternative therapeutic potential for AD whose treatment measure does not exist at present.”
Discussion
Line 176 -> "Previous studies, this study..." -> review and correct;
Response: The paragraph including the sentence was revised accordingly (Lines 239-240).
Line 188 -> "While Although BV..." -> correct;
Response: The phrase was corrected to “While BV has been…” (Line 246).
Line 192 -> "...the Aβ accumulation, Accumulation..." -> correct;
Response: It was corrected to “the Aβ accumulation…” (Line 265).
Line 223 -> "components(s)" -> correct;
Response: It was corrected to “component(s)” (Line 298).